# Differentiating Cyclability and Kinetics of Na⁺ Ions in Surface-Functionalized and Nanostructured Graphite Using Electrochemical Impedance Spectroscopy

Sonjoy Dey  and Gurpreet Singh *

Department of Mechanical and Nuclear Engineering, Kansas State University, Manhattan, KS 66506, USA; sonjoy@ksu.edu
* Correspondence: gurpreet@ksu.edu

**Abstract:** The revolution in lithium-ion battery (LIB) technology was partly due to the invention of graphite as a robust negative electrode material. However, equivalent negative electrode materials for complementary sodium ion battery (NIB) technologies are yet to be commercialized due to sluggish reaction kinetics, phase instability, and low energy density originating from the larger size of Na⁺-ion. Therefore, in search of the next-generation electrode materials for NIBs, we first analyze the failure of graphite during reversible Na⁺ ion storage. Building upon that, we suggest surface-functionalized and nanostructured forms of analogous carbon allotropes for enhancing Na+ ion storage. During long-term rigorous cycling conditions, Graphene Oxide (GO) and Graphene nanoplatelets (GNP) exhibit higher Na⁺ ion storage (157 mAh g⁻¹ and 50 mAh g⁻¹ after 60 cycles, respectively) compared to graphite (27 mAh g⁻¹). Optimizing alternative NIBs requires a comprehensive analysis of cycling behavior and kinetic information. Therefore, in this investigation, we further examine ex-situ electrochemical impedance spectroscopy (EIS) at progressive cycles and correlate capacity degradation with impedance arising from the electrolyte, solid electrolyte interphase formation, and charge transfer.

**Keywords:** graphite; GO; GNP; electrode kinetics; sodium ion battery; rechargeable batteries; EIS; NIB



## 1. Introduction

Over the past three decades, LIBs have brought revolutionary achievements as a power source for portable electronic devices and electric vehicles. Nevertheless, the uneven distribution of Li metal in the earth's crust has necessitated the search for an inexhaustible alternative material and technology [1–3]. Because of the large abundance and uniform distribution in the earth crust, low extraction, and purification costs of sodium ores, NIB is considered viable successor to LIB, particularly for large-scale or stationary energy storage. However, regardless of the advantages mentioned above, the incorporation of NIBs in modern applications is held back by sluggish reaction kinetics, phase instability, low energy density, and undesirable irreversibility—mainly resulting from the large ionic size of Na-ion (1.02 Å) [4–6]. Moreover, as the research advancement in LIB technology has progressed satisfactorily [7–9], superior negative electrodes have been proposed for LIBs [10–12]. Be that as it may, satisfactory advancements in negative electrodes for NIBs are yet to be found [12–14].

The comprehensive utilization of graphite as a negative electrode material due to its low cost, high capacity, relatively long cycle life, and processing feasibility facilitated the commercial applicability of LIBs [15,16]. All the same, graphite cannot be utilized as feasible negative electrode material for NIBs due to its inability to host Na⁺ ions in the interlayers (0.335 nm), minimal incorporation of Na⁺ ions in the graphite host (equivalent to the development of ~NaC₁₈₆), and no energetic driving force encouraging Na⁺ ions to intercalate, resulting in deposition of Na⁺ ions on the surface of the electrode [15,17].

Moreover, the usage of graphite in NIBs yields distinctive solid electrolyte interphase (SEI) layers with unexpected precipitates resulting in low initial coulombic efficiency and large consumption of electrolytes [18]. Although there have been rigorous research advancements in identifying suitable negative electrode materials for NIBs, most of them include the usage of polymers, oxides, sulfides, carbonaceous materials, and hybridized structures—suffering from low cycling stability, poor thermal stabilities, inferior electrical conductivities, significant volume expansion, low energy density for utilization of heavy metals, and complicated synthesis procedures increasing the cost of fabrication [19].

Among various strategies for energy storage enhancement, heteroatom doping is an effective strategy for structural modifications. Previous investigations report that the enhancement in Na$^+$ ion storage in the modified structure is mainly attributed to the electron conjugation effect, enlargement of interlayer spacing, construction of new covalent bonds, and escalated binding affinity of Na$^+$ ions [20–22]. Nanostructure utilization is another attractive approach in the augmentation of Na$^+$ ion storage properties, which captivates the scientific community with various conveniences. Amelioration in nanotechnology has enabled the implementation of nanostructures as battery electrodes due to clear advantages regarding high rate capability, power density, higher alkali metal ion solubility, higher gravimetric capacity, reduced memory effect, superior fracture toughness, and fatigue resistance. Specifically, smaller particle size enhances the alkali metal ion migration and reduces the miscibility gap, due to which the kinetics and thermodynamics of active materials improve [23]. Although nanostructured electrodes contain a more significant number of active sites for hosting alkali-metal ions, the process leading up to SEI formation in carbonate-based electrolytes is unavoidable—causing higher irreversible charge losses [24,25].

In this work, three carbon allotropes were studied in a Na$^+$ ion half-cell setup to evaluate their electrochemical properties. Specifically, for a lucid comparison of electrochemical properties, traditional graphite was studied with functional groups containing GO and nanostructured GNP. Microscopic and spectroscopic studies laid the grounds for the subtle structural differences between the materials. Na$^+$ ion storage within carbon allotropes with minute discrepancies was evaluated using galvanostatic charge-discharge and stable cycling conditions to procure the investigation's primary objective. Ex-situ EIS investigation shed light on the capacity decaying phenomena in detail for Graphene with functional groups and nanostructured states during Na$^+$ ion storage and compared with graphite, the robust negative electrode material for LIB, NIBs counterpart. Furthermore, impedance arising from the electrolyte, the SEI formation, and particle separation due to charge transfer were dissected from EIS spectra, highlighting the reason for graphite's low-capacity contribution and the subsequent higher capacity gain in surface functionalized GO and nanostructured graphite. This study, therefore, highlights the preeminence of surface functionalization and nanostructuring while investigating superior low-cost negative electrode materials for NIBs.

## 2. Materials, Methods, and Characterization

### 2.1. Materials

Graphite and GNP were purchased from the vendors for this experiment. graphite was purchased from Aldrich Chemistry, Burlington, MA, USA. The GNPs were purchased from Asbury Carbons Inc., Asbury, NJ, USA. graphite and GNP were used for electrode preparation as-received without further modification and the surface area value was 50–80 m$^2$ g$^{-1}$ and 250 m$^2$ g$^{-1}$, respectively reported by manufacturers. GO was prepared by the modified hummers method depicted in a previous study [26]. The typical surface area of GO prepared via the modified hummers method is approx. 8 m$^2$ g$^{-1}$ [27]. The coin cell components were purchased from MTI Corporation, Richmond, CA, USA. Finally, the ultra-high purity Argon gas for assembling the cells was provided by Matheson, Manhattan, KS, USA. Carbon Black and Polyvinylidene difluoride (PVDF) were purchased from Alfa Aesar,

Haverhill, MA, USA. The N-methyl-2-pyrrolidinone (NMP) was purchased from Sigma Aldrich, St. Louis, MO, USA.

### 2.2. Electrode Preparation

The active materials such as graphite powder, GNP, and GO (70 wt.%) were mixed with carbon black (15 wt.%), which worked as a conducting agent, and PVDF (15 wt.%), which functioned as a binding material. Then, one by one drop of NMP was added gradually until a homogenous paste was obtained. With this mixture, a film with a homogeneous thickness of ~125 μm was cast onto a copper foil with a thickness of 9 μm using the Blade coating technique. Finally, the substrate was dried overnight at 80 °C to evaporate the solvent and achieve electrodes for half-cell assembly.

### 2.3. Cell Assembly

The dried electrodes were punched out using a 7.94 mm (radius) circular punch and used as a working electrode in the coin-like (CR2032) sodium-ion half-cell setup where pure sodium metal was used as reference and counter electrodes having a diameter of 14.3 mm and a thickness of 75 μm. The electrolyte solution was a mixture of 1 M sodium perchlorate ($NaClO_4$) (Alfa Aesar) in 1:1 $v/v$ ethylene carbonate (EC): dimethyl carbonate (DMC). A glass separator with a diameter of 19 mm and a thickness of 25 μm separated the two electrodes being pre-soaked with the electrolyte. The half-cells were assembled in an inert atmosphere utilizing an ultra-high pure Ar-filled glovebox and further tested in 100 mA g$^{-1}$ constant current density in an Arbin BT2000 multichannel battery tester.

### 2.4. Characterization Methods

Phenom Pure G6 Desktop SEM was utilized to obtain scanning electron microscopy (SEM) images. Transmission electron microscopy (TEM) images were obtained using a Phillips CM100 instrument under an accelerating voltage of 100 kV. The active materials for making the electrodes were characterized via Raman Spectroscopy and X-ray photoelectron spectroscopy (XPS). Raman spectra were collected using a He-Ne laser (wavelength of 633 nm and power of 17 mW) on a confocal Raman imaging system named Horiba Jobin Yvon LabRam Aramis. On the other hand, in this study, the XPS spectra were obtained using XPS, PHI Quantera SXM using monochromatic Al-Kα with an energy of 1486.6 eV. XRD reflection was obtained using a PANalytical Empyrean multipurpose X-ray diffractometer with a scan rate of 1° minute$^{-1}$.

### 2.5. Electrochemical Impedance Spectroscopy

Electrochemical impedance spectroscopy, an influential tool for investigating material properties and electrode reactions, was utilized to study the carbon allotropes' kinetic behavior. The frequency sweep across large magnitudes helps users to differentiate features and signatures at different timescales. In a half-cell, when the electrode comes in contact with a nonaqueous electrolyte, it forms surface layers with electrolyte solvent and salt molecules. Rapid electrode material change occurs during consequent sodiation/desodiation, which causes an extensive morphological change of active material. Furthermore, charge transfer occurs on the current collector/electrode or the current collector/film interface, which helps to complete the electrochemical redox reaction. Based on these physicochemical processes, plausible electrical equivalent circuit models can be considered for investigating impedances arising from the electrode material, electrolyte, and current collector [28–30]. The most critical parameters utilized in this study are discussed below:

- The ohmic resistance of the cell comprises ionic and electronic resistances from the electrodes, flow fields, current collectors, and contact resistances. As the impedance of a resistor includes only the real part of a complex number, the resistances arising from different physicochemical processes are readily observable from the Nyquist plot.
- The non-ideal electrode surface analysis is conducted using a constant phase element (CPE), and the impedance arising can be formulated as follows:

$$Z_{CPE}(\Omega) = \frac{1}{Y_0(j\omega)^N} \tag{1}$$

where $j = \sqrt{-1}$, $\omega$ = frequency and the significance of $N$ is illustrated in the later section.

- Inductive behavior at high frequency is observed due to electron movement in the potentiostat cables, and impedance arising can be determined from the following equation:

$$Z_L(\Omega) = j\omega L \tag{2}$$

where $L$ = inductance in $\Omega \cdot$S.

- The diffusional transport of electroactive species can be denoted by Warburg impedance, determined by the equation below:

$$Z_T(\Omega) = \frac{\mathrm{R coth}(\tau j\omega)^P}{(\tau j\omega)^P} \tag{3}$$

where $\tau$ contains information regarding diffusion constant.

This study obtained electrochemical impedance spectra in a 0.01–100 Hz range using a CH instrument potentiostat after every 1st, 11th, 21st, 31st, 41st, 51st, 61st, and 71st cycle sodiation. In addition, all the data for EIS were obtained after relaxing the cell at 0.1 V after sodiation.

## 3. Results and Discussion

### 3.1. Microscopic Analysis

Figure S1a–c shows the SEM micrographs of graphite, GO, and GNP material, respectively. While graphite possessed the largest flake size compared to the other two materials (Figure S1a), the GO sample demonstrated a fairly solid flake-like morphology, not peeled into individual sheets (shown in the inset of Figure S1b) [31]. The SEM images of the GNP sample demonstrated the smallest flake size with rough surface morphology, displayed on the inset of Figure S1c [32]. Figure 1a–i show the TEM images and Selected Area Diffraction Patterns (SAED) of the graphite, GO, and GNP samples. Figure 1a,b show graphite's low-resolution and high-magnification TEM images. The particle size (length) of the graphite flake observed in the TEM images was 5.7 µm. The bright spots are arranged ring-like in Figure 1c, providing information regarding the polycrystallinity of the graphite flake on the inset. 002 and 004 crystal planes were clearly identified from the SAED spot pattern of graphite flake, in accordance with previous investigations [33,34]. Figure 1d,e. depict the low-resolution and magnified images of GO material. The TEM micrograph of the GO sheet illustrated rippling, folding, and scrolling features. In addition, the agglomeration effect is also observed within the GO sheets, which might inhibit the accessibility of electrolytes within individual sheets [35].

Furthermore, Figure 1f depicts the SAED pattern of the GO material along with the individual image of the sheet from which the pattern was obtained, shown on the inset. The bright hexagonal spots appearing in a ring-like fashion depict the presence of a few graphene sheets in the spot from which the SAED pattern was taken [36]. The d-spacing of the 002 plane for GO was higher than graphite (4.02 Å, compared to 3.3 Å of graphite) and in accordance with previous investigations [37,38]. The smaller sizes of the individual graphite sheets in the GNP sample (~500 nm) are visible in Figure 1g,h—compared to graphite flakes. The SAED pattern obtained from GNP, illustrated in Figure 1i, describe light halos along with hexagonal spot pattern with different intensities characteristic of GNP sheets [39]. The 002 crystal plane with a d-spacing of 3.4 Å was observed for the GNP, quite similar to graphite [40,41].

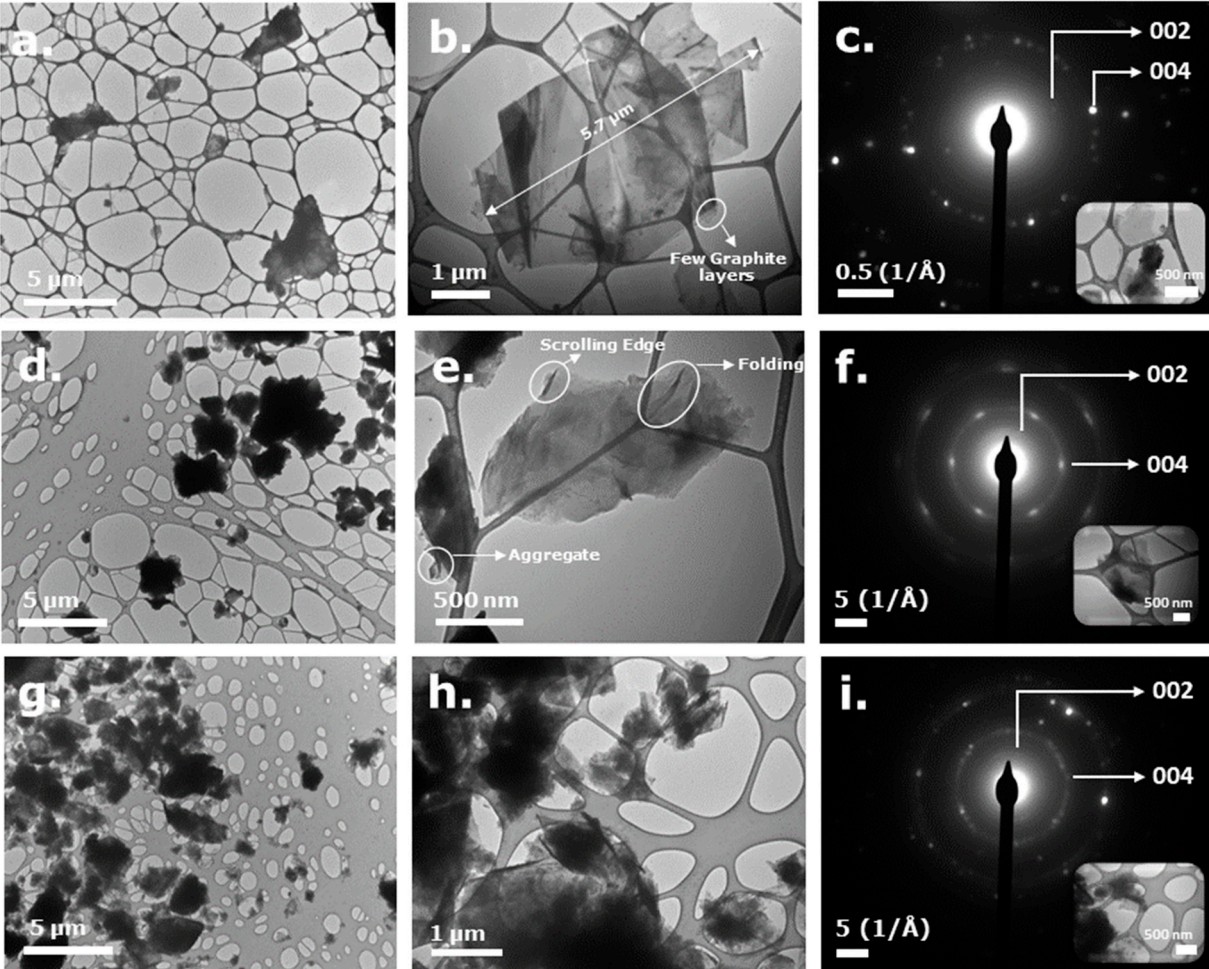

**Figure 1.** TEM micrographs of (**a**–**c**) graphite; (**d**–**f**) GO; (**g**–**i**) GNP illustrating a de-magnified and magnified view of different and individual flakes, respectively, and corresponding SAED patterns. The TEM images on the SAED pictures' inset represent the spots from which the diffraction patterns were taken.

*3.2. Spectroscopic Analysis*

The structural characterization of the materials in this study involved using this ubiquitous Raman spectroscopic technique which yields a wealth of information regarding the material's morphology. Figure 2a–c illustrate the Raman spectra obtained from the graphite powder, exfoliated GNP, and GO powder. From Figure 2a, it is apparent that the G peak of graphite powder appears at 1586 cm$^{-1}$. The reason for the visibility of the G peak is due to the stretching of the sp$^2$ bonds [42]. On the other hand, the D peak for the graphite sample is observed at around 1342 cm$^{-1}$. The D peak arises from the symmetry of the A$^{1g}$ symmetric phonons [43]. The 2D peak, an overtone of the D peak, is also seen in the graphite sample, situated at twice the peak position of the D peak. As, in perfect Graphene or graphite sheet, the D peak disappears, it can be implemented that the sample here possesses some defective sites [44]. The D and G bands appeared at 1329 cm$^{-1}$ and 1584 cm$^{-1}$ for the GNP material. Another peak between the D and the G peak seemed at 1457 cm$^{-1}$, which can be termed as the D″ peak originating from the fraction of amorphous carbon contained within the sample—can be produced by the presence of graphene layers [45].

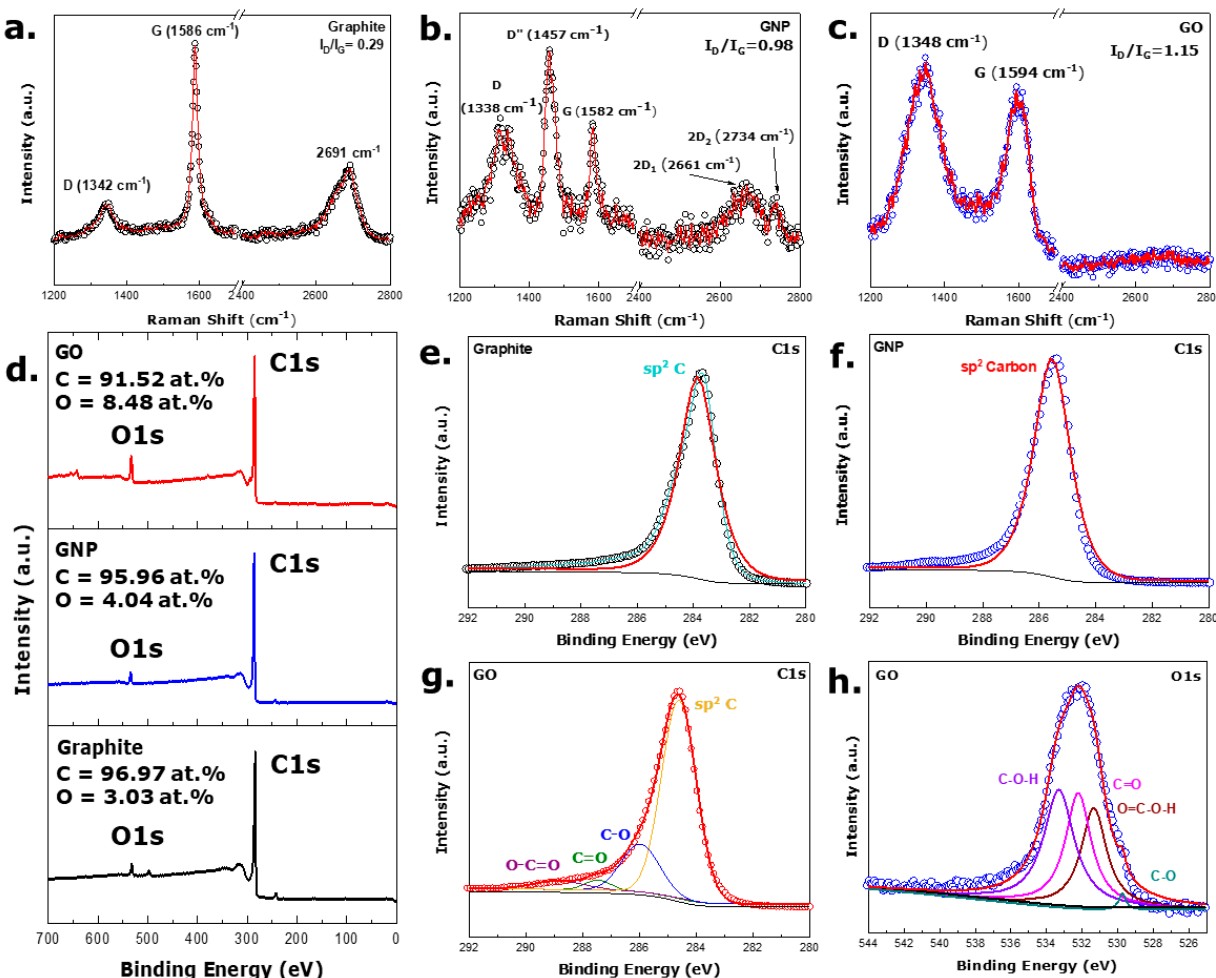

**Figure 2.** Raman spectra analysis of (**a**) graphite; (**b**) GNP; (**c**) GO electrode illustrating the structural configuration within; (**d**) XPS survey scan of graphite, GO, and GNP materials illustrating the elements present within; High-resolution C 1s XPS spectra of (**e**) graphite, (**f**) GNP, (**g**) GO material; O 1s spectra of (**h**) GO material.

Moreover, the 2D band has two peaks, $2D_1$ at 2661 cm$^{-1}$ and a low-energy shoulder $2D_2$ at 2732 cm$^{-1}$. The splitting of the 2D band originates from the dispersion energies of the pi-electron splitting—emerging from the interaction between the neighboring graphitic planes. The splitting of the 2D band confirms the formation of few-layered Graphene as the splitting is observed only in a perfectly A-B stacked few-layer Graphene, by about 5–6 layers [43,46]. For the GO sample, the D and G bands were located at 1348 cm$^{-1}$ and 1594 cm$^{-1}$, respectively, with a flat 2D profile. The graphite sample's lowest $I_D/I_G$ value of 0.29 indicates a lower number of defective sites present than the other materials. As the increase in the ratio of $I_D$ and $I_G$ represents more defects in the material, the GNP sample is suggestive of more defective sites than any other material, and the GO sample is second in the order. A slight shift in the G band is observed in the GO sample, which is thought to have originated from oxidation [47]. Moreover, as the average size of the sp$^2$ carbon domains gets reduced and more edge defects are induced, the intensity ratio value is lower than that of pure Graphene. As per expectation, no pronounced 2D peak was visible in the GO sample.

XPS provides valuable information about different samples, which enhances our understanding of functionalized groups present within the material and helps us evaluate chemical species, chemical states, and bonding. Figure 2d shows the XPS survey scan of the three active materials used to make sodium-ion half cells. The survey scan indicates

that the C1s peak is the most prominent in all the samples. Considering the area under each peak and making ratios with the total area under the peak, the elemental analysis was conducted, presented in the table. The survey scan resembles the elemental analysis, as all the samples had over 90% carbon. In addition, the maximum amount of oxygen was found in the GO sample, which is as expected, and the oxygen in other samples is negligible.

The deconvolution of high-resolution spectra of the samples further validates the data obtained from the survey scan. From the deconvolution of the graphite (Figure 2e) and GNP (Figure 2f) samples' C1s spectra, only a sharp peak for the sp$^2$ carbon at $284.6 \pm 0.1$ eV was apparent. As the oxygen percentage was very little, even the deconvolution of the peaks did not result in significant oxygen functionalization for graphite and GO [48–50]. From the high-resolution C1s spectra of GO, oxygen functionalities are evident in the material (Figure 2g). The deconvolution of the high-resolution spectra reveals C=C, C-O, C=O, and O=C-OH bonds present in the region of $284.6 \pm 0.1$ eV, $286.8 \pm 0.1$ eV, $288.2 \pm 0.1$ eV, and $289.4 \pm 0.1$ eV, respectively. The bonds mentioned above resemble well with the previous literature [51,52]. The deconvolution of high-resolution O 1s spectra depicted C-O, O=C-O-H, C=O, and C-O-H bonds at $529.4 \pm 0.1$ eV, $531.34 \pm 0.1$ eV, $532.21 \pm 0.1$ eV, $533.33 \pm 0.1$ eV, respectively from Figure 2h for the GO sample [35,53].

### 3.3. Reflection Analysis

Figure S2 illustrates the XRD reflections from GNP, GO, and graphite. graphite demonstrated the sharpest peak among the three samples at $2\theta = 26.74°$, representing a (002) diffraction line with interlayer d spacing of 3.33 Å in the crystal [54]. The most prominent peak for the GO material resided at $2\theta = 25.89°$ with a somewhat broad feature, symbolizing (002) plane with a d-spacing of 3.49 Å, higher than graphite—much closer to single Graphene flake thickness [55]. Previous literature ascribed the change in interlayer distance due to the absorption of water molecules in the basal planes and existing structural defects within the material [56]. Small peaks are also observed at $2\theta = 28.08°$ and 33.3°, indicating that GO might not fully connect with oxygen atoms. The appearance of the (002) peak of GNP is much broader and less intense than graphite, residing at $2\theta = 26.97°$, with a d-spacing of 3.3 Å, much similar to graphite [57]. The as-obtained d-spacings from the XRD technique are in accordance with the findings from the SAED pattern.

### 3.4. Electrochemical Analysis

Figure 3a–c show the galvanostatic charge-discharge (GCD) for graphite, GO. GNP electrodes assembled in a Na-ion half-cell setup at a current density of 100 mA g$^{-1}$ when cycled in the 0.01–2.5 V voltage window. The first cycle charge capacity for the graphite, GO. GNP electrodes were about 31, 191, and 54 mAh g$^{-1}$ with a coulombic efficiency of 15.46%, 15.39%, and 15.23%, respectively. The low coulombic efficiency of the first cycle can be attributed to the phenomena of solid electrolyte interphase (SEI) formation, where sodium ions and electrolyte species react with the carbon electrode surface and thus form an electronically insulating but ionically conducting layer. Several other reasons contributing to the low initial coulombic efficiency include the co-intercalation of solvated ions originating from the solvent molecules of electrolytes and the irreversible storage of larger Na$^+$ ions into different sites [58]. For the GNP material, the irreversible capacity in the first cycle can also be attributed to high specific surface area—resulting in increased exposure to electrolyte molecules and subsequent parasitic reactions [25]. For the GO, apart from the SEI formation, the presence of oxygen functional groups also leads to parasitic reactions with Na$^+$ ions [59]. The results of this reaction are visible in the voltage range of 0.2 and 1 V [17]. Differential capacity curves derived from the GCD curves in Figure 3d–f show that the three reduction peaks at 0.05 V, 0.5–0.7 V, and 0.7–0.9 V are visible for the graphite and the GNP electrode. The GO electrode broad peak in 0.01–0.56 V is observed. As these peaks are not visible in any of the electrodes during the 2nd cycle onwards, they can be assigned to forming the SEI layer [60,61]. The broadest peak observed in the case of the GO electrode is further highlighted in Section 3.5. Again, the plateau at the low voltage region

(0.01 V; still visible after the first cycle) can also be assigned to the insertion and extraction of $Na^+$ ions into or from the microcrystallites [61,62]. The unavailability of the plateau discussed previously in the case of the GO electrode can be ascribed to the variation in the crystallinity of the sample. In addition, the availability of a peak around the 0.8 V region indicated the interaction of $Na^+$ ions with oxygen-containing functional groups (broadly visible in the case of the GO electrode with the highest amount of O presence validated from the XPS investigation). The desodiation peak around ~0.08 V visual for all three electrodes can be ascribed to the adsorption of $Na^+$ ions into the defective sites present within the materials [63]. Noticeably, some noisy features were discerned from Figure 3e. Previous literature has addressed several reasons for noise generation in differential capacity curves. The noise in voltage results from thermal noise (Johnson-Nyquist noise) and current noise originating from the power source that imposes an additional noise in the voltage. Several other effects, such as a change in open circuit voltage as a result of the temperature gradient, the voltage drop in internal resistance due to a variation of constant source current, change in ambient temperature, and aging of cells, have also been reported responsible for noisy features in differential capacity profiles [64]. Methods for reducing generated noise include simple data reduction, moving averages, and FFT smoothing. Fitting the as-obtained voltage profile with different Gaussian processes has also been proposed for high-quality differential capacity curves [65]. As the noise observed in Figure 3e did not overshadow any valuable features, no curve fitting, filtering, or smoothening operation was performed. Figure 3g shows the cycling stability plot for the three electrodes in a Na-ion half-cell setup. Figure 3g shows that the coulombic efficiency of the graphite electrode became stable at 60% after the first 11 cycles.

In contrast, one of the GNP electrodes overtook graphite from the second cycle onwards. On the consecutive cycles, the coulombic efficiency of the GNP electrode achieved stability of 75% and stayed at the same number for the rest of the cycles. On the contrary, the coulombic efficiency of the GO electrode showed a better contribution than the rest of the carbon-based electrodes, scoring around 85% at the 12th cycle, gradually rising, and reaching 93% at the 61st cycle. Furthermore, a capacity of 178 mAh g$^{-1}$ was achieved even after 50 cycles from the GO electrode. The comparatively high capacity of GO can also be attributed to oxygen functional groups, which expand the interlayers of graphite. Previous studies have indicated that oxygen-containing groups in the interlayer of graphite drop down the energy barrier for $Na^+$ insertion to 0.053 eV [66]. The high capacity of the GO electrode verifies the claim mentioned above.

### 3.5. Electrochemical Impedance Spectra Analysis

To better understand the behavior of the three-carbon allotropes in an SIB half-cell, EIS measurements were taken after discharging the cells as the carbon allotropes were used as working electrodes in a half-cell setup. Furthermore, complete relaxation was ensured for all the cells by obtaining the data after the open-circuit voltage of the cells reached 0.1 V. Therefore, the EIS spectra were found without any disturbances or anomalies. The fitting process was carried out for quantitative analysis of the EIS data, representing the respective electrodes' main physical processes in the frequency range measured. The fitting data obtained using the Zview software (v 3.2b, Copyright Scribner Associates, Inc., Southern Pines, NC, USA) is illustrated in Table S1. The Nyquist plots of the graphite, GO, and GNP cells are shown in Figure 4a–c up to the 71st cycle discharge. The main features of the spectra can be evaluated based on the following phenomena:

- The inductive behavior observed in the high-frequency range of all the spectra can be primarily ascribed to the measurement system (wires connecting the half-cell to the potentiostat) or due to cell geometry or cell windings. Although, as in all spectra, the inductive loop is visible, in this study, it is assigned to be arising from the measurement system. The high-frequency intercept of the real impedance axis with the inductive circle corresponds to the sum of internal ohmic resistance, including the electrolyte, active material, current collectors, and electrical contacts [67].

- A prominent depressed semicircle appears in the medium frequency range. It is fitted using an $R|Q$ element in the equivalent circuit. It denotes resistance arising from the solid electrolyte interface (SEI) formation (i.e., the desolvation of $Na^+$ ions and their incorporation into the SEI). Specifically, this semicircle is absent in the first cycle. Still, it progresses throughout the cycling conditions as the formation of SEI layers passivates the anodes and prevents delamination of individual layers while leading to continuous capacity decay [68]. The significance of SEI growth is illustrated further in the following section.

- Another small flat semicircle, which reflects the charge transfer resistance and interfacial capacitance fitted with an $R|Q$ element, appears in the low-frequency region. As the electrodes utilized in this investigation are neither ultrathin nor thick, a moderate semicircle is observed in the case of all three carbon allotropes. This semicircle can again be correlated to the time constant of the charge transfer being coupled with a double-layer type interfacial capacitance ($C_{dl}$). Several other factors, such as interparticle electronic resistance and porosity of the electrode, may influence the diameter of the semicircle further [69].

- A sloping line or tail at very low frequencies indicates $Na^+$ ion diffusion in the active material of the cell electrodes. The Warburg element ($W$) fits the tail at the low-frequency region, establishing a connection to the mid-frequency responses modeled with $R|Q$ elements. In this study, the fitting is conducted, placing the Warburg element in series with the double layer capacitance as the impedance arising due to SEI formation is well separated from the charge transfer process, and impedance contribution from various processes can be well separated in this way [70]. Previous literature has modeled this tail generalizing the Warburg element as a $Q$ or $CPE$ with $N$ different from $-0.5$. However, in the Nyquist plots obtained, the tail inclination is higher than $45°$, which a CPE generalization cannot define. This non-ideal behavior can be ascribed to the anisotropic diffusion in the particles with variable size distribution [71,72]—which is more prominent in the case of the GNP electrode as the electrode was prepared using sonication.

As stated before, the high-frequency inductive loop and the intercept of the impedance curve with the real axis represent the ohmic resistance $R_{ohm}$, otherwise known as the static electrolyte resistance. The gradual increase in $R_{ohm}$ value can be observed in Figure 4d for all three electrodes. The highest $R_{ohm}$ value was observed in the case of the graphite electrode in comparison with the GO and GNP electrodes. Although no clear comparative trend in the case of $R_{ohm}$ value for three electrodes was found in Figure 4d, a general rise in $R_{ohm}$ value as a function of cycle numbers was observed. Previous investigations indicate that the increase in electrolyte resistance is a result of the decrease in the number of active sites approachable to $Na^+$ ions, decomposition of electrolyte substances ($ClO_4^-$ and EC: DMC), or an increase in the concentration of organic species in electrolyte solution—restricting the motion of $Na^+$ ions [73,74]. The highest electrolyte resistance of graphite electrodes indicates that using graphite as an electrode in $Na^+$ ion cells might lead to higher electrolyte decomposition and no storage conditions.

The mid-frequency depressed semicircle can be attributed to the impedance arising from the formation of the SEI layer. The semicircle is fitted using a parallel $CPE$ and resistance ($R_{SEI}$) due to its non-ideal behavior (Figure 4e, equivalent circuit shown in Figure S3 [75–77]). The trend in the impedance increase due to SEI formation is illustrated in Figure 4e, where the highest gradual increase in $R_{SEI}$ value can be ascribed to the GNP material. Lower to higher $R_{SEI}$ values in subsequent cycles can be attributed to the formation of the SEI layer in the first cycle and then the proceeding of SEI growth during the prolonged cycling process. In ideal conditions, electrolyte decomposition during the first cycle should help construct a 'protective layer' covering the electrode's surface, saving it from further degradation. Unfortunately, as the SEI layers are permeable to $Na^+$ ions and impermeable to other electrolyte components and electrons, they should protect the charged electrode from further corrosion. However, as a stable SEI layer is not formed, and

an increase in SEI resistance is observed, a possibility of byproduct generation (gaseous phases) remains. Although $NaClO_4$-based electrolyte is known for thicker and more rigid SEI formation, the option of byproduct generation due to repetitive SEI formation cannot be fully eliminated [78]. The byproducts of the SEI include NaCl generation from the degradation of $NaClO_4$, carbonate-based molecules (primarily present in the inner part of the electrode), and CO-based species [79]. In parallel to SEI growth, sodium corrosion in the carbon might occur, resulting in capacity fading due to the system's loss of usable $Na^+$ ions.

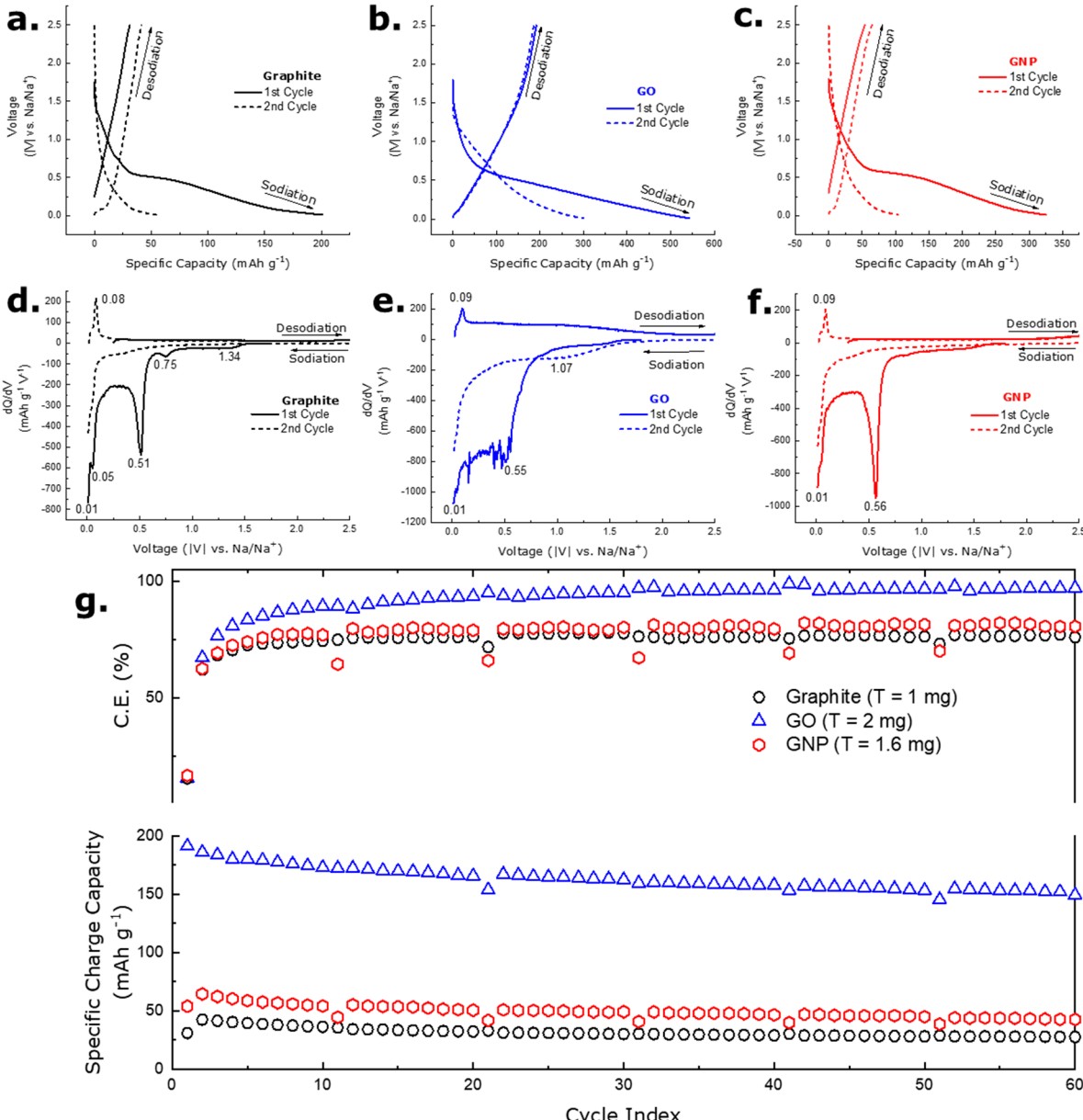

**Figure 3.** GCD plots for (**a**) graphite; (**b**) GO; (**c**) GNP electrode; Differential capacity curves derived from GCD curves for (**d**) graphite; (**e**) GO; (**f**) GNP electrode; (**g**) specific charge capacity and subsequent coulombic efficiency of graphite, GO, and GNP electrodes as a function of cycle numbers at a constant 100 mAg$^{-1}$ current density. T-value indicates the total mass of the electrodes containing an active material, conductive agent, and binder.

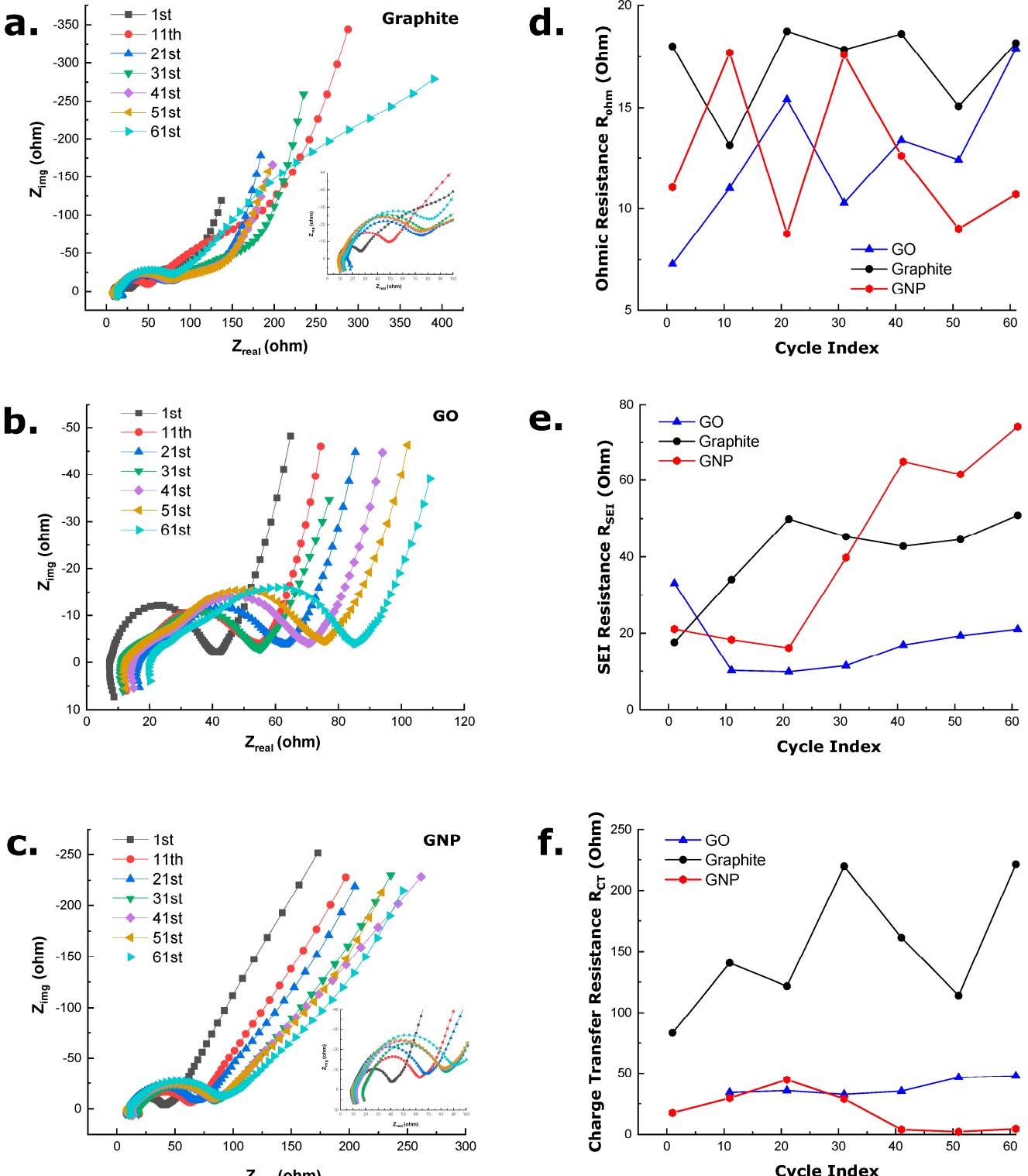

**Figure 4.** Real impedance vs. Imaginary impedance (Nyquist) plot as a function of cycle number for (**a**) graphite; (**b**) GO; (**c**) GNP electrodes; comparative analysis of (**d**) ohmic resistance; (**e**) SEI resistance; (**f**) charge-transfer resistance as a function of cycling progression for graphite, GO and GNP electrodes.

Furthermore, due to SEI growth, contact loss between the carbon particles, current collector/carbon, binder/carbon, and binder/present happening inside the electrode might increase the $R_{SEI}$ value [80]. Although all the EIS spectra were collected after the sodiation

process, the resistance due to SEI formation and evolution was found since SEI coverage using NaClO$_4$ salt is more prominent in the case of Na$^+$ ion half-cells compared to their Li$^+$ ion counterparts [79]. The high $R_{SEI}$ value of the GNP electrode (41st cycle onwards and higher than the GO electrode) stems from the increased surface area of nanoparticles. As high surface area cultivates more sites for Na$^+$ ion interaction, the possibility of continuous SEI formation and degradation on individual sheets up to prolonged cycling conditions might exist, leading to constant decomposition of electrolyte, an indirect result of which is a very high coulombic efficiency [24,25]. Additionally, in previous studies, it was found that very little sodium can be incorporated into the graphite host (equivalent to the development of ~NaC$_{186}$). As there is no energetic driving force to encourage Na$^+$ ions to intercalate, the metal instead just gets deposited on the surface of the electrode. As a result, the impedance should rise, and the semicircles just became more prominent on the consecutive cycles. Thus, the capacity obtained for the graphite electrode can be thought of as mainly originating from the carbon black, which was used as a conductive agent while assembling the cell. Although thinner SEI is believed to be formed on the surface of GO when compared with the GNP electrode [81], the first cycle $R_{SEI}$ value of the GO electrode was higher than the rest, which can be correlated to the broad peak appearing in the differential capacity curve (Figure 3e) occurring due to Na$^+$ ion interaction with oxygen functional groups.

Mid-frequency second semicircle present in Nyquist plots for all cycles in the graphite electrode and from the 11th cycle onwards for the GO and GNP electrodes can be attributed to the charge transfer phenomena and fitted with another resistor ($R_{CT}$) and capacitor-like effect $CPE_{dl}$ in parallel and shown in Figure S3. Specifically, the number of electrochemical reactions to facilitate the intercalation/deintercalation processes are identified by the charge transfer phenomenon (an electron from the outer circuit traveling through the current collector and depositing on the electrode surface). As cycling proceeds and the electrical load gets applied, various charge carriers will inherently stockpile on the electrode/electrolyte surface and establish ions deposition in the active material [82]. From the fitted results obtained and listed in Table S1, a trend in the value of *N* in $CPE_{dl}$ is observed where $0.3 < CPE\text{-}N < 0.9$ for GNP, $CPE\text{-}N > 0.7$ for the GO, and graphite electrode. As the value of *N* approaching near 1 indicates a porous electrode surface's capacitor-like behavior, the Na$^+$ ion storage in graphite and GNP is more likely to be deposited on the edges of the graphite sheet [83]. Previous investigations have shown that GO with greater defect density enhances the capacity of Na$^+$ ion storage [84]. The charge transfer resistance ($R_{CT}$) values obtained after fitting are tabulated (in Table S1), and a plot showing a comparison between the $R_{CT}$ values for three different electrodes is depicted in Figure 4f. The highest $R_{CT}$ values are attributed to the graphite electrode, while the GNP electrode's $R_{CT}$ value is higher than the GO electrode. The highest $R_{CT}$ value of the graphite electrode can be linked with the formation energies of small amounts of sodium-rich-binary-graphitic-intercalation compounds (Na-rich b-GICs). The Nobuhura et al. investigation indicates the relation between the formation energy of b-GIC with the stretching and subsequent destabilization of the C-C bonds within the graphene sheets during cycling—which might cause such a high $R_{CT}$ value of graphite [85]. Similar $R_{CT}$ values of GNP electrodes also make sense as GNP electrodes contain nano-graphite sheets dispersed within. As the GO electrode comprises oxygen functional groups, the effect mentioned above might be minimized as surface functional groups have been shown to promote Na storage [22]. Further, reasons for the increase in $R_{CT}$ value for all three electrodes can be correlated to the aging mechanism of the cells as the EIS spectra were obtained at an increment of cycles. Specifically, as aging continues in the cell, an increase in $R_{CT}$ indicates a lower amount of Na$^+$ ions participating in redox activity with individual graphene sheets. Additionally, morphological changes of the electrode, including particle cracking, pore clogging and particle disconnection due to repeated cycling maybe a cause of increase in $R_{CT}$ value of all three electrodes [82,86].

In an ideal situation, the tail of the Nyquist plot at the low-frequency region is expressed by finite space type Warburg impedance mirroring solid state diffusion [87]. In a perfect case, the *W-P* element fitted should be 0.5, indicating only solid-state diffusion is

taking place. As the fitted data obtained and shown in this study for all three electrodes show a *W-P* value of ~0.35, it is, therefore, assumable that several other processes during the mass transport are responsible, including migration and liquid state diffusion. Previous literature indicates that the deviation of the tail region from 45° inclination is mainly caused by interference from electrolyte phase diffusion. Therefore, even if the diffusion coefficient is determined using this data, it would resemble the ion diffusion process in both solid and electrolyte phases and thus is avoided in this study [88–90]. It is to be noted that the diffusion constant can be calculated from the *W-τ* value for finite diffusion using the following equation:

$$W - \tau = \frac{L^2}{D} \tag{4}$$

*L* is the diffusion length of cations in the electrode, and *D* is the chemical diffusivity [91]. On the other hand, the *W-R* value indicates the diffusion resistance and a notable trend is observable. The meager *W-R* value in graphite electrodes can be correlated to the capacity achieved from these electrodes. The very high *W-R* value of the graphite electrode indicates that some diffusion happens during the first cycle. For that, a high resistance (*W-R* value) is seen as $Na^+$ ion diffusion in graphite is unfavorable. In the successive cycles, no capacity is achievable from the graphite electrode, and therefore no diffusion of ions happening within the solid state occurs. In this case, the diffusion of ions might be purely attributed to liquid phase diffusion and, thus, the deposition of some $Na^+$ ions on the surface of the top layer of the electrode. Surprisingly, GO with surface modification groups shows good capacity around 170 mAhg$^{-1}$ even after the 60th cycle, which can be attributed to the diffusion of $Na^+$ ions into some solid state, particularly oxygen functional groups. Among the three electrodes investigated here, GO possessed the lowest *W-R* value with the highest capacity—indicating the dominance of solid-state diffusion. However, the GNP electrodes' highest *W-R* value is seen, which can also be correlated to their capacity. It is to be noted that the size of the particles used as active material plays a vital role in capacity. Previous studies have illustrated the benefit of nanoparticles (comparable to GNP material used in this study) in the case of fast-charging capability, higher solid solubility, and gravimetric capacity. As a capacity of ~50 mAh g$^{-1}$ is achievable from the GNP electrode even after 60 cycles, the high Warburg resistance might arise from the solid-state diffusion and the liquid-state diffusion (due to the irregularity of the 45° tail). Although nanomaterials provide additional sites for $Na^+$ ion diffusion and reduce the diffusion length to some extent, resistances might still occur due to the non-uniformity of the active material size within the electrode [23].

## 4. Conclusions

In summary, this work featured three similar carbon allotropes with very subtle disparities as working electrodes in $Na^+$ ion half cells. Microscopic techniques (TEM) and spectroscopic techniques, i.e., Raman and XPS, elucidated the minute differences between the materials where GO had the highest oxygen functionalities and GNP had the highest defective sites. Furthermore, the $Na^+$ ion storage within the three materials was compared, where graphite's contribution to $Na^+$ ion storage was the least (27 mAh g$^{-1}$ at 60th cycle), while GO (157 mAh g$^{-1}$) being the highest followed by GNP (50 mAh g$^{-1}$). Furthermore, the gradual capacity degradation in graphite, GO, and GNP was analyzed by EIS, where impedance arising due to electrolyte, SEI formation, and charge transfer was highlighted in depth. graphite and GNP's low capacity and subsequent capacity decay were correlated with high impedance arising from SEI formation and subsequent segregate formation. Additionally, higher charge transfer resistance was observed in graphite and GNP, indirectly indicating electrode morphological change and particle separation during prolonged cycling conditions. However, GO with surface functional groups exhibited lower SEI and charge transfer resistance, indicating the formation of facile intercalation sites for $Na^+$ ions.

Moreover, the capacitive storing of Na$^+$ ions was elucidated from the *CPE* value of graphite and GNP, demonstrating that in the case of nanostructures and neat graphitic material, Na$^+$ ions prefer to be stored on the surfaces. This study correlates the Na$^+$ ion storage with kinetic information obtainable from EIS spectra, elucidating the efficiency of two pronounced methods of material modification (presence of surface functional groups and nanostructure creation) for fabricating next-generation electrodes utilizing low-cost materials. In addition, this study showed the effectiveness of the EIS technique in differentiating three carbon allotropes with minute disparities.

**Supplementary Materials:** The following supporting information can be downloaded at: https://www.mdpi.com/article/10.3390/batteries9110534/s1, Figure S1: SEM micrograph of (a) graphite, (b) GO, (c) GNP material; insets contain an enlarged view of flakes; Figure S2: XRD reflection of graphite, GO, and GNP; Figure S3: Equivalent circuit model used for fitting the EIS spectra of the three electrodes. No charge transfer resistance was observed for the GO electrode in the first cycle; Table S1: Fitting results from EIS experiments.

**Author Contributions:** Conceptualization: G.S.; methodology: G.S. and S.D.; analysis: S.D.; investigation: S.D.; resources: G.S.; data curation: S.D.; writing—original draft preparation: S.D.; writing—review and editing: G.S.; visualization: S.D.; supervision: G.S.; funding acquisition: G.S. All authors have read and agreed to the published version of the manuscript.

**Funding:** This work is supported by the National Science Foundation Grant #1743701, 1454151, and 2025298.

**Data Availability Statement:** The data presented in this study are available at https://ksuemailprod-my.sharepoint.com/:f:/g/personal/sonjoy_ksu_edu/EnqFYs4Daw1Om5CEID0Cyq0BMxHDWsi3cK0nB0zLp5iLjQ?e=5Itlji (accessed on 31 August 2023).

**Acknowledgments:** This work is supported by the National Science Foundation grants #1743701, 1454151, and 2025298. The research was performed in part in the Nebraska Nanoscale Facility: National Nanotechnology Coordinated Infrastructure and the Nebraska Center for Materials and Nanoscience (and/or NERCF), which are supported by the National Science Foundation under Award ECCS: 2025298, and the Nebraska Research Initiative. The authors also acknowledge the help of Shakir Bin Mujib regarding SEM and XRD analysis. The publication of this article was financed with support from the Kansas State University Open Access Publishing Fund.

**Conflicts of Interest:** The authors declare no conflict of interest. The funders had no role in the design of the study; in the collection, analyses, or interpretation of data; in the writing of the manuscript; or in the decision to publish the results.

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
