# Peer review of "Differentiating Cyclability and Kinetics of Na+ Ions in Surface-Functionalized and Nanostructured Graphite Using Electrochemical Impedance Spectroscopy"

_batteries, doi:10.3390/batteries9110534_

Round 1
Reviewer 1 Report
Comments and Suggestions for Authors
The manuscript entitled "Differentiating cyclability and kinetics of Na+ ions in surface-functionalized and nanostructured graphite using Electrochemical Impedance Spectroscopy" reported the application of carbon-derived materials graphene oxide (GO), graphene nanoplatelets (GNP), and graphite as anodes for sodium-ion batteries. For physical properties, the materials were examined with TEM, Raman, and XPS. For electrochemical properties, the anode materials were investigated in the half-cell system by using CV, galvanostatic charge/discharging methods, and impedance spectroscopy. The results indicated that GO anode material showed a higher specific capacity of 170 mAh g-1 compared with that of DNP (~ 50 mAh g-1) and graphite (~ 40 mAh g-1). Such a higher capacity of GO could be attributed to the lower SEI layers, charge transfer, and diffusion resistance compared to those of others. However, there are several weaknesses that require the attention of the authors. Because of these points, this manuscript can only be published in Batteries after several major revisions. The specifications are mentioned as follows:
- The surface area and pore size distributions of the materials should be added to the manuscript because the porosity of the materials may affect the impedances. The XRD and SEM should be additional provided in the manuscript.
- Is there any previous publication using the same equivalent circuit model as the author suggested in Figure S1? If it’s a new model for the first time, how could the author make it?
- The author should make a sodium-ion full-cell for the GO materials.
English is ok.
Reviewer 2 Report
Comments and Suggestions for Authors
The authors provided a reasonable scientific analysis on the surface modification of Graphite for Na-ion batteries. The quality and the amount of data and analysis is good. However, before accepting in MDPI Batteries, the following comments must be addressed.
1. The authors do not provide the affiliation properly. Please change.
2. Although the authors mentioned in the title that the paper was focused on the EIS analysis of the surface modification of graphite, the main content was actually not mainly focused on EIS. Therefore, the authors need to modify the title to avoid the confusion.
3. In the TEM images (Figure 1c, 1f and 1i), the authors need to index different diffraction planes in each given electron diffraction patterns.
4. In Figure 4a-4c, the authors are encouraged to mention the potential mechanism on why in the first cycle, the irreversible capacity in Figure 4a and Figure 4c is much less than that in Figure 4b. What happens? I do not think it is solely due to the SEI formation, given the fact that all the three cells have a SEI formation CV peak at 0.55 V according to the authors' analysis.
5. In Figure 4e, why do the authors have some "noisy" feature between 0.01 and 0.55 V? Was that due to the cell construction? Need to explain and add details in the manuscript.
6. In Figure 4b, the positive imaginary part of impedance has little information, just plotted the EIS data from "0" for both x-axis and y-axis. In addition, if we compare Figure 4b and Figure 4c, the interfacial resistance seems to be the same. However, this quite similar interfacial resistance led to a huge difference in performance (Figure 3g). The authors need to add detailed explanation about it in the manuscript in the proper location.
7. As the authors mentioned graphite electrode for batteries, the following publications should be cited in the introduction part: a. Energy Storage Materials, 57, 429-459, 2023; b. Nano Research Energy, 2023, 2: e9120081.
Round 2
Reviewer 1 Report
Comments and Suggestions for Authors
The author gave sufficient answers to all the questions, and the manuscript has also been revised. I recommend this revised manuscript be published in Batteries.